# Research on Stability Evaluation of Perilous Rock on Soil Slope Based on Natural Vibration Frequency

**Yanchang Jia** [1,*]**, Guihao Song** [1,*]**, Luqi Wang** [2]**, Tong Jiang** [1]**, Jindi Zhao** [1] **and Zhanhui Li** [1]

1    College of Geosciences and Engineering, North China University of Water Resources and Electric Power, Zhengzhou 450046, China
2    School of Civil Engineering, Chongqing University, Chongqing 400030, China
*    Correspondence: jiayanchang@ncwu.edu.cn (Y.J.); songgh0322@163.com (G.S.); Tel.: +86-18737132511 (Y.J.); +86-18436919556 (G.S.)

**Abstract:** Perilous rock instability on the soil slope brings a substantial threat to project operation and even people's lives. The buried depth of the perilous rock is a challenge to deal with and primarily determines its stability, and the indirect rapid identification of its buried depth is the key to its stability evaluation. The paper aims to find a new and quick method to measure the buried depth of perilous rock on the soil slope and to solve the hard-to-measure buried depth stability evaluation. When the damping ratio is less than one, and the deformation is linear elastic throughout the amplitude range, the potentially perilous rock vibration model may reduce to a multi-degree-of-freedom vibration one. By theoretical deduction, a quantitative relationship is established among the perilous rock mass, the basement response coefficient, the buried depth of the perilous rock, and the natural horizontal vibration frequency. In addition, the accuracy of this relationship is confirmed via numerous indoor experiments, showing that the horizontal vibration frequency of the perilous rock model in one dimension increases as the buried depth increases. Finally, based on the natural vibration frequency and guided by the limit balance model, a stability evaluation model of the perilous rock on the soil slope is constructed. Hence, the example shows that the method is feasible. The research findings are of vital significance for the stability evaluation of the perilous rock on the soil slope and give a novel approach and theoretical foundation for quick identification and monitoring.

**Keywords:** perilous rock on soil slope; vibration frequency; stability evaluation; buried depth of perilous rock; perilous rock model

## 1. Introduction

Perilous rocks on high and steep slopes are unstable, particularly when exposed to environmental or anthropogenic forces, such as precipitation, earthquake, or vibration [1–4]. The safety coefficient of the perilous rock stability on the soil slope refers to the ratio of the anti-overturning torque to the overturning one. When the ratio is greater than one, the slope is stable; when it is equal to one, the slope is in the limit equilibrium state; when it is less than one, the slope is damaged, and the perilous rock becomes unsteady. Then, under the action of gravity, these unstable rocks roll rapidly and seriously affect the project operation and even people's safety. In China, perilous rock is one of the common geological disasters in mountainous areas and along roads. Even though the rock mass involved in a perilous rock collapse is moderate, it features wide distribution, sudden eruption, fast speed, high energy, long movement distance, and difficulty in monitoring and early warning. In addition, it poses a serious threat to people's safety [5–9]. Especially with the implementation and advancement of the western development strategy and the "Belt and Road" initiative, many large-scale infrastructures are being constructed or have been built in southwestern China and along the Silk Road Economic Belt, where the risk of collapses and rockfalls may occur. For example, a building of the Sichuan–Tibet Railway will face

engineering geological problems: the collapse of the shallow surface of the slope and the perilous rocks during construction and operation, due to the steep landform along the line, the complex geological environment conditions, and the development of unfavorable geological conditions, such as active fault zones and high ground stress [10,11]. It can be said that collapses and rockfalls are an unavoidable and important challenge in rock slope engineering, and scientific researchers and engineering technicians have paid much attention to it in recent years [12,13].

The buried depth of the perilous rock on the soil slope is an important parameter to evaluate its stability, which is hard to measure directly by instruments, so it brings great challenges for its identification, stability evaluation, and monitoring and early warning. To solve this problem, this paper, through theoretical deduction, model tests, and field tests, constructs a method based on dynamic characteristic parameters to identify the buried depth of perilous rock on the soil slope and then achieves its stability evaluation with the limit equilibrium theory. Research into the best way to assess the safety of a rock on the soil slope involves breaking the process down into qualitative stability assessments and quantitative numerical modeling approaches. It is difficult to achieve a quantitative evaluation of the stability of perilous rock by using qualitative research methods. The stability of the perilous rock may be quantitatively evaluated using the limit equilibrium and numerical simulation methods; however, the primary elements influencing the steadiness of perilous rock, such as its buried depth, size of the bond area, and bond strength of the perilous rock structures, are hard to grasp. Owing to current scientific and technological advancements, a technique has evolved for gauging the stability of the perilous rock based on its variation of dynamic characteristics. Amitrano et al. [14] analyzed the vibration before the collapse of the rock mass on the coast of Normandy in western France and concluded that vibration characteristics, such as vibration amplitude, can play a certain role in early monitoring and forewarning of a potentially fatal rockfall; Got et al. [15] concluded based on the displacement and vibration of a natural limestone cliff rock mass in the southeastern part of Vickers Mountains that spectrum analysis can be used as a supplement to the displacement velocity to provide early warning of perilous rock collapse.

Based on the vibration monitoring technology, Bottelin et al. [16] studied the dynamic response of four unstable rock masses in the Alps, and according to the measurements, there was a distinct energy peak at the resonance frequency of the perilous rock mass's vibration spectrum; Ma et al. [17], by comparing the safety factors of the test blocks, found that the vibration characteristic parameters, such as frequency, have a certain indication effect on the stability of the rock block. After analyzing the mechanical indicators, such as friction force, it was concluded that rock mass safety monitoring based on the vibration mode will play an important role in practical engineering [18–20]; then, Jia Yanchang et al. [21–27] obtained the change rule of the dynamic characteristic parameters of perilous rock with structural damage through tests and a theoretical analysis and realized the quantitative stability evaluation based on the vibration frequency. Numerous investigations have shown that for the stability assessment of the rock blocks it is reliable to use dynamic indicators. When it comes to collapse catastrophes, this is one of the most useful technological tools available. Taking perilous rock on the soil slope as the research object, the paper establishes a quantitative link among the rolling rock mass, the foundation response coefficient, and the buried depth of the perilous rock. Significant progress has been made in determining the natural vibration frequency in the horizontal direction theoretically and computing the buried depth of the perilous rock rapidly. Laboratory experiments confirm the accuracy of the theoretical model. Based on the natural vibration frequency, the stability evaluation model of the perilous rock on the soil slope is then established in conjunction with the limit equilibrium model, and the calculation example demonstrates the feasibility of this approach. The research findings are extremely significant for checking the precariousness of rocks on a slope and provide a novel approach and theoretical foundation for the quick identification and monitoring of perilous rocks.

## 2. Materials and Methods

### 2.1. Perilous Rock on Soil Slope and Research Method

A perilous rock appears when a mass of rock on a soil slope splits from its parent body but is still on the slope (see Figure 1), and it may cause a collapse disaster at any time due to gravity or other natural factors, such as rainfall, earthquake, wind action, and animal interference. After the perilous rock collapses, the rockfall often moves downward at a high speed along the slope, with an extremely powerful force of impact. The incidence of perilous rock collapse is connected to factors such as the external force, the nature of the slope on which the perilous rock is located, the rock's proclivity, and its buried depth. The perilous rock vibration model is simplified into a multi-freedom vibration model. The quantitative relationship among perilous rock mass, rock bottom response coefficient, perilous rock buried depth, and natural horizontal vibration frequency is established by theoretical inference. In the experiment, several perilous rock models with different masses and different buried depths were set up, and the vibration frequencies under the same buried depths and the buried depths under the same vibration frequencies were compared, researching the relationship between the buried depths of the perilous rock and the vibration frequencies.

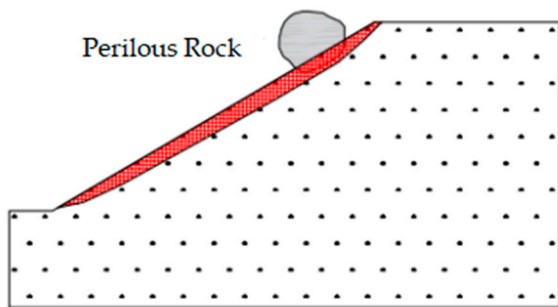

**Figure 1.** Schematic diagram of perilous rock on a soil slope.

### 2.2. Dynamic Characteristic Model of Perilous Rock on Soil Slope

Perilous rock formations will shake when prompted by external forces. In the case of the homogeneous isotropic foundation soil, when the damping ratio is smaller than one, the multi-degree-of-freedom rock model simplifies down to a single-degree-of-freedom model. Rock vibration models may be simplified by considering just a few degrees of freedom due to the linear elastic character of the deformation over the whole amplitude range. See Figure 2.

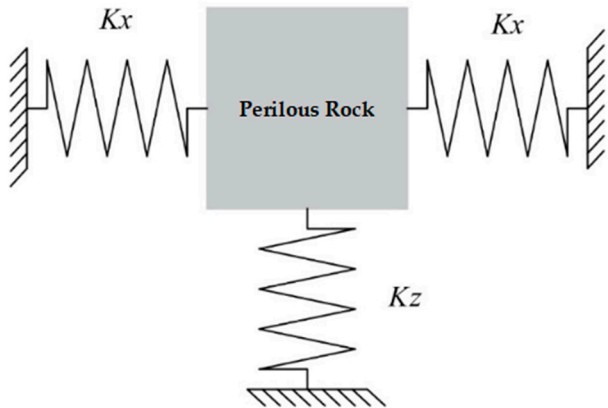

**Figure 2.** Dynamic characteristic model of perilous rock.

According to dynamic theory, when an outside force causes the rolling rock slope to vibrate, the vibrations travel parallel to the slope. The potentially fatal chasm between the

rock and the starting point is initially fixed at x; the horizontal stiffness of the soil layer is $K_x$, and according to Hooke's law, the perilous rock is subjected to a force $K_x x$ moving in the opposite direction of the shift, calculated by using Newton's second law and the equations of horizontal mechanical equilibrium for the potentially perilous rock:

$$M\ddot{x} = -K_x x \tag{1}$$

$$p^2 = K_x/M \tag{2}$$

The frequency of the rock's natural vibrations, without any dampening, is denoted by P, and its mass, M, is the other key variable. Then, Equation (3) was applied for the free vibration of the perilous rock on the soil slope without any damping.

$$\ddot{x} + p^2 x = 0 \tag{3}$$

We let the equation at time t be solved as:

$$x = e^{xt} \tag{4}$$

We replaced the solution with Equation (3) to obtain the roots of the equations and identified equations:

$$s_{12} = \pm ip \tag{5}$$

$$s^2 + p^2 = 0 \tag{6}$$

Combined with the above equations, the equation's universal answer is:

$$x = C_1 e^{ipt} + C_2 e^{-ipt} \tag{7}$$

We applied Euler's formula to rewrite Formula (7):

$$x = C\cos pt + D\sin pt \tag{8}$$

The magnitudes of the integral constants C and D were defined by the beginning circumstances. By substituting $t = 0$, $x = x_0$, $\dot{x} = \dot{x}_0$ into Formula (8), we obtained the value of C and D as:

$$C = x_0, D = \dot{x}_0/p \tag{9}$$

$$x = x_p \cos pt + \sin pt\dot{x}/p \tag{10}$$

The trigonometric transformation of Formula (10) resulted in the following new form:

$$x = A\sin(pt + \alpha) \tag{11}$$

A is a constant, where:

$$A = \sqrt{x_0^2 + \left(\frac{\dot{x}_0}{p}\right)^2}, \alpha = tg^{-1}\frac{px_0}{\dot{x}_0} \tag{12}$$

The system's vibration period is:

$$T = 2\pi/p = 2\pi\sqrt{M/K_x} \tag{13}$$

The system's vibration frequency is:

$$f = 1/T = \frac{1}{2\pi}\sqrt{\frac{K_x}{M}} \tag{14}$$

The relationship between the stiffness coefficient of the foundation soil $K_x$ and the foundation reaction force coefficient k is:

$$K_x = khb \tag{15}$$

Equation (16) represents the horizontal vibration frequency at which a potentially perilous rock is most likely to oscillate after combining Equations (14) and (15):

$$f = \frac{1}{2\pi}\sqrt{\frac{khb}{M}} \tag{16}$$

Considering the ratio effect of the height of the perilous rock to its breadth parallel to the direction of vibration at its fundamental frequency, it becomes clear that the perilous rock on the soil slope operates against the damping force when the damping is weak. Over time, the amplitude of the vibrations will diminish as the system's mechanical energy is transformed into heat and eventually lost as heat. The resulting Formula (16) can be expressed as:

$$f_d = \frac{\sqrt{1-\xi^2}}{2\pi}\sqrt{\frac{(\frac{a}{H_s})^2}{(\frac{a}{H_s})^2+1}}\sqrt{\frac{khb}{M}} \tag{17}$$

where $f_d$ is the damped system's fundamental vibration frequency (Hz); $\xi$ is the damping ratio of the system (unitless); a is the perilous rock's dimension in the direction of the vibration, m; b is the perilous rock length in the vertical vibrational axis, m; M is the rock's bulk, kg; h is the perilous rock's buried depth, m; H is the height of the precarious rock, m; $H_s$ is the vertical distance from base to top of perilous rock, m.

To further verify the correctness of Formula (17), the following part verifies whether the relationship among the buried depth, mass, and vibration frequency of the perilous rock is consistent with Formula (17) through the model test.

## 3. Model Test

*3.1. Test Equipment*

(1)  Vibration meter

Specifically, a Microchip Cube wireless vibrometer manufactured by Beijing Beike Andi Technology Development Co., Ltd. (Beijing, China) was used in the experiment with a quick sampling rate, a high signal-to-noise ratio, and wireless data transmission. The frequency of the object's natural vibrations in all three dimensions was determined using the Fourier transform. The exact vibration meter parameters are depicted in Table 1.

**Table 1.** The performance indexes of Microchip Cube.

| Parameter | Minimum | Maximum | Unit |
|---|---|---|---|
| range | −5.0 | +5.0 | g |
| sampling ratio | 5.0 | 800 | Hz |
| resolution | 0.50 | — | µg |

(2)  Vibration measurement

We attached the vibration detector to the model's summit and then used epoxy glue to connect the detector's fixed plate to the perilous rock block shown in the foreground. The installation of the layout sensor occurred after the epoxy resin had dried and the test model had been taken entirely. The velocity/acceleration in the three dimensions was activated after the hammer excited the wide frequency domain, measuring the object's time-domain representation in Figure 3. After obtaining a representation of the observed object's frequency domain by filtering and the Fourier transform (Figure 4), the measured object's natural vibration frequency could be determined.

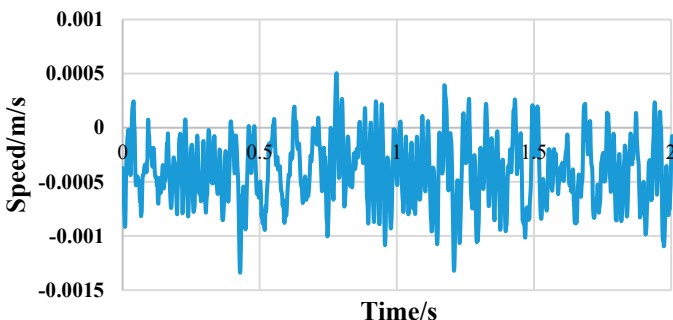

**Figure 3.** Vibration time-domain diagram.

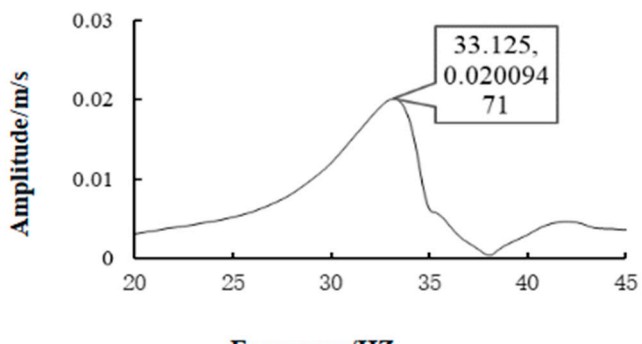

**Figure 4.** Vibration frequency-domain diagram.

### 3.2. Experimental Model

(1) Test model

In the experiment, cubes of different sizes and qualities were used to simulate perilous rocks. The dimensions of models A, B, and C were 0.3 m × 0.3 m × 0.4 m, 0.3 m × 0.3 m × 0.3 m, and 0.3 m × 0.3 m × 0.2 m, respectively, and the masses were 84.3 kg, 63.2 kg, and 42.5 kg, respectively. Figure 5 depicts the experiment's perilous rock model.

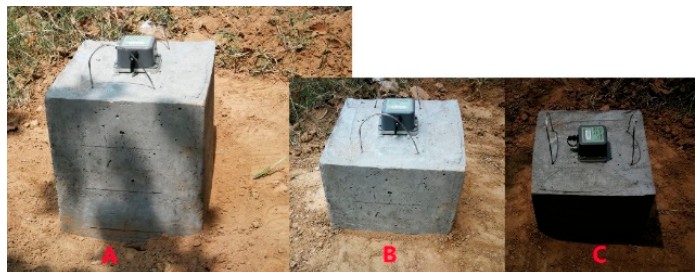

**Figure 5.** Perilous rock test models.

(2) Foundation soil

We took the soil on the open-top landslide body on the Dadu River National Road slope as the test foundation soil. The natural density, natural dry density, deformation modulus, and grain group content of the soil were obtained through conventional soil tests. Table 2 displays the standard properties of the foundation soil.

### 3.3. Test Plan

The models were buried on the soil slope to simulate perilous rocks, and 1/5, 2/5, 3/5, and 4/5 of the model's height governed the buried depth. The slope angle θ remained unchanged. We used a hammer with a wide frequency range to hit the model, and then a vibrometer. The acceleration time–history curve of the perilous rock model was

obtained. We obtained the model's power spectrum by first applying a horizontal filter to the acceleration time–history curve and afterward by performing a Fourier transform on the filtered data. The test plan is displayed in Figure 6.

**Table 2.** Foundation soil parameters.

| Test Factor | Average | Unit |
|---|---|---|
| natural density ($\gamma$) | 1.92 | g/cm$^3$ |
| internal friction angle ($\varphi$) | 20.0 | $^\circ$ |
| coefficient of subgrade reaction (k) | $6.5 \times 104$ | kN/m$^3$ |
| cohesion (C) | 7.0 | kN/m$^2$ |

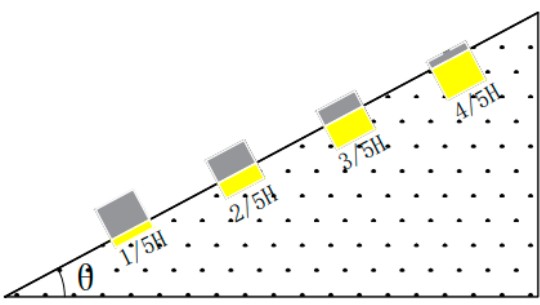

**Figure 6.** Test scheme.

### 3.4. Test Results

After the test, perilous rock models with various buried depths had their horizontal natural vibration frequencies measured. Table 3 displays the results of our statistical analysis.

**Table 3.** Experimental block horizontal vibration frequency.

| Models | Buried Depth/cm | Measured Frequency/Hz | Buried Depth Calculation/cm | Error/% |
|---|---|---|---|---|
| A | 8 | 13.6 | 7.49 | 6.37% |
| A | 16 | 23.2 | 16.72 | 4.51% |
| A | 24 | 31.8 | 24.61 | 2.52% |
| A | 32 | 39.2 | 31.18 | 2.56% |
| B | 6 | 16.5 | 6.34 | 5.69% |
| B | 12 | 25.3 | 12.36 | 3.03% |
| B | 18 | 32.6 | 17.51 | 2.73% |
| B | 24 | 39.9 | 23.51 | 2.02% |
| C | 4 | 17.7 | 3.84 | 3.92% |
| C | 8 | 27.4 | 8.32 | 3.97% |
| C | 12 | 35.4 | 12.82 | 6.83% |
| C | 16 | 40.9 | 16.26 | 1.63% |

The test findings reveal a maximum error of 6.83%, a minimum error of 1.63%, and an average error of 3.82% for the inferred buried depth of the perilous rocks.

### 3.5. Discussion

Figure 7 displays the experimentally determined connection among the buried depth of perilous rock, the vibration frequency, and the mass of perilous rock.

(1) Correlation between the buried depth of the perilous rock and the natural vibration frequency.

Figure 7 shows that the natural vibration frequency of the perilous rock grows nonlinearly as the buried depth increases.

By transforming Equation (17), the relationship between the buried depth of the perilous rock and the natural vibration frequency could be obtained as:

$$h = \frac{4\pi^2 M f_d^2 (a/H_s^2 + 1)}{kb(1 - \xi^2)(a/H_s)^2} \tag{18}$$

Equation (18) demonstrates that as the buried depth of the perilous rock increases, the frequency of its natural vibrations grows steadily in a quadratic fashion. Additionally, the theoretical model has an average inaccuracy of 3.82% when comparing the experimentally observed buried depth of the perilous rock model to the predicted buried depth, showing that there is a strong theoretical relationship between the frequency of the natural vibrations and their buried depth, as proposed in this work. Unlike the manual method to measure the buried depth of the perilous rock, this paper, by measuring its natural vibration frequency, provides the theoretical basis and new technical support for its early identification, stability evaluation, and monitoring and early warning.

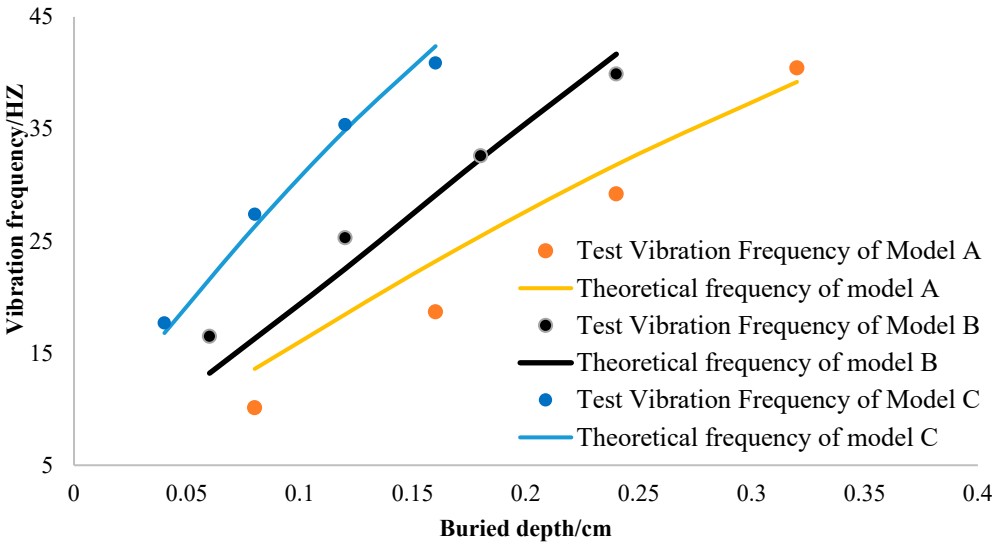

**Figure 7.** Relationship between natural vibration frequency and buried depth.

(2)    Relationship between the natural vibration frequency and the perilous rock mass.

Figure 7 shows that, for a given natural vibration frequency, a lighter model has a shallower buried depth than a heavier one. When the models have the same natural vibration frequency, the buried depths of the models fall in the following order: A > B > C. If two rocks of equal mass are buried at the same depth, the vibration frequency of a perilous rock with a larger mass is lower than that with a smaller mass. The order of the natural vibration frequency of the models under the same buried depth is C > B > A.

Equation (17) leads to the following conclusion on the link between the perilous rock's bulk and its inherent vibration frequency:

$$M = \frac{kbh(1 - \xi^2)(a/H_s)^2}{4\pi^2 f_d^2 (a/H_s^2 + 1)} \tag{19}$$

The natural vibration frequency reduces quadratically with the enhancing mass of the perilous rock, as shown by Formula (19). Data gathered from experiments involving the magnitude of perilous rocks and the frequency of natural vibrations confirm the accuracy of this result.

## 4. Perilous Rock Stability Evaluation

### 4.1. Force Analysis of Perilous Rock

Figure 8 depicts the results of a force study performed on the potentially perilous rock, where F is the friction force between the perilous rock and the soil, N; G is the gravity of the perilous rock, N; $E_p$ is the passive earth pressure, N; $E_a$ is the active earth pressure, N; $F_e$ is the external force of the perilous rock, N; $\theta$ is the angle of the slope, $°$; Q is the water pressure at the bottom, N; z is the depth of the calculation point from the soil surface, m; $\delta$ is the friction angle between the perilous rock and the soil, where the value is $\varphi$, $°$.

$$E_p = 0.5\gamma h^2 \tan^2(45° + \varphi/2)b \tag{20}$$

$$E_a = 0.5\gamma(h - z)^2 \tan^2(45° - \varphi/2)b \tag{21}$$

$$z = 2c/(\gamma \tan(45° - \varphi/2)) \tag{22}$$

$$F = G\sin\theta + F_e\cos\theta + E_a - E_p \tag{23}$$

For rolling stones in a rolling state:

$$M_{qf} = \left(\frac{H}{2} - h\right)G\sin\theta + \frac{a}{2}Q + \left(\frac{H}{2} - h\right)F_e\cos\theta + \frac{a}{2}F_e\sin\theta + \frac{1}{3}E_p h^2 \tag{24}$$

$$M_{kqf} = \frac{a}{2}G\cos\theta + \frac{1}{3}E_p h^2 \tag{25}$$

Among them, $M_{qf}$ is the rolling motion state rolling rock overturning moment, and $M_{kqf}$ is the rolling motion state rolling rock anti-overturning moment.

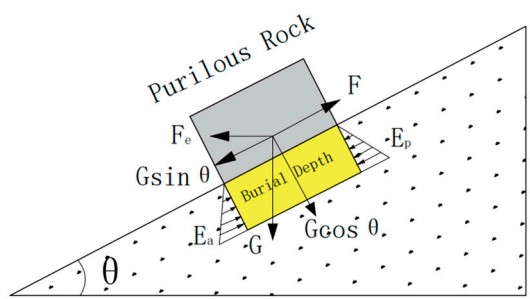

**Figure 8.** Force diagram of perilous rock.

### 4.2. Stability Evaluation Model

According to the limit equilibrium model, the safety factor K of the rolling rock under the condition of translational motion was obtained:

$$K = \frac{E_p + F + (ab + 2ah + 2bh)c}{E_a + G\sin\theta} \tag{26}$$

The safety factor K of the rolling stone under the condition of rolling motion state:

$$K = \frac{M_{kqf}}{M_{qf}} \tag{27}$$

Formulas (18), (20)–(23) were put into Formula (26), and Formulas (24) and (25) were put into Formula (27) to obtain the connection between the safety factor of the rolling stones and the natural vibration frequency under various motion states.

### 4.3. Example

During the impoundment of the Monkey Rock Reservoir in Ganzi Prefecture, Sichuan Province, there were obvious signs of deformation on the outer high retaining wall and

the inner slope of the highway at the gully downstream of the open-top landslide. These deformations severely threaten people's safety and property within the S211 Provincial Highway and the downstream reservoir area. According to the site geological survey, the open-top landslide is about 14.5 km away from the Monkey Rock Dam and is located at the right bank of Dadu River in Kairao Village, Gezong Township, Danba County, and about 450 m upstream of Xihegou. The deformation body is about 460 m long along the river, with a top elevation of 2080 m, a bottom elevation of 1820 m or less, and a total volume of about four million cubic meters. The shallow surface of the open-top landslide is composed of colluvium–deluvial large boulders, block stones, and gravelly soil. The underlying surface is Silurian greenschist, containing phyllite. There is an interlayer compression fracture zone in the rock layer, and the surface cracks are mainly distributed in the middle and rear of the deformation body. The maximum boulder size is 40 m × 10 m × 15 m, and there are many cracks on the side surface, extending to the bottom of the boulder. The total volume of the boulders is about 150,000 cubic meters.

It is vital to evaluate the safety status of perilous rocks on the soil slope and issue an early warning. The main aspect of determining the stability of the perilous rock is to determine its buried depth. However, the shallow surface of the open-top landslide has complex terrain and a harsh geological environment, also with many unknown dangerous factors. Therefore, it is extremely dangerous to carry out manual operation, adding difficulties to directly measuring the buried depth of the perilous rock. The traditional surveying method is limited; hence, it is urgent to find a more practical measurement method to conform to the project's reality.

This paper establishes the evaluation model of the perilous rock on the soil slope. Based on the natural vibration frequency, the buried depth of the perilous rock can be determined quickly and efficiently, and its safety state can be evaluated. Firstly, the natural vibration frequency and the volume of perilous rocks in the horizontal direction were measured, and their vibration model was simplified into a multi-degree-of-freedom vibration model. Secondly, the buried depth was calculated according to the relationship between the natural vibration frequency of the perilous rock and its buried depth in the Formula (17). Finally, the safety factor was further determined according to the equilibrium conditions, and the safety state of the perilous rock was evaluated. Table 4 shows the parameter values of the two perilous rocks (see Figure 9) on the open-top landslide on the bank slope of the Dadu River Reservoir.

**Table 4.** Parameter indexes of the perilous rocks.

| Length of Perilous Rock | /cm | Width/cm | Height/cm | Mass/kg | Slope Angle/° |
|:---:|:---:|:---:|:---:|:---:|:---:|
| A | 3.33 | 5.21 | 7.23 | 327,386.50 | 25.00 |
| B | 2.85 | 4.89 | 4.56 | 165,866.65 | 25.00 |

The average monthly precipitation in the Dadu River Basin is 100–700 mm, and it reaches the maximum in July and August at nearly 700 mm. The basic earthquake intensity is VII degrees. On-site monitoring data show that the natural vibration frequency of perilous rock A is 3.95 Hz and that of perilous rock B is 4.28 Hz. The natural density of the foundation soil is 1.92 g/cm$^3$; the internal friction angle is 20.0°; the foundation reaction force coefficient is $6.5 \times 10^4$ kN/m$^3$, and the cohesion is 7.0k N/m$^2$. We substituted the parameters of perilous rock A and B into Formula (18) to calculate that the buried depth of perilous rock A is 2.18 m and that of perilous rock B is 1.05 m. According to Formula (26), the safety factor of perilous rock A is 1.11, and that of the rolling rock B is 1.11 under the condition of translational motion. According to Formula (27), the safety factor of rolling rock A is 2.33 and that of rolling rock B is 2.83 under rolling motion. Both perilous rocks are in a very stable condition.

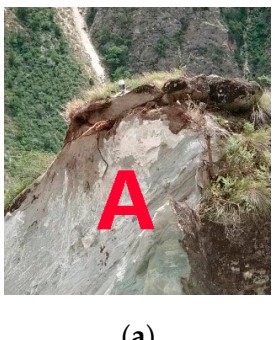
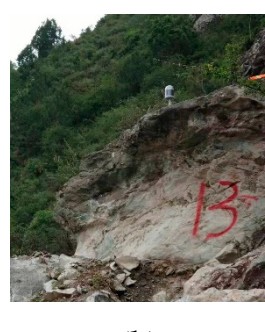

| (a) | (b) |

**Figure 9.** Perilous rocks. (**a**) A is a perilous rock on the open-top landslide; (**b**) B is a perilous rock on the open-top landslide near A.

### 4.4. Research Prospects

In the alpine canyon areas of southwestern China, a lot of perilous rocks exist near railways and highways. The government attaches great importance to investigating and cleaning them, with massive governance in progress. The research results are being piloted and popularized in the Monkey Rock and Baihetan Hydropower Station. The research is in the early stage. The next step is to take the remote non-contact laser vibration measuring instrument to replace the current contact wireless sensor. The perilous rock areas are difficult to reach by manpower in high mountain and canyon areas. Therefore, the remote non-contact laser vibration measuring instrument can be used to survey the natural vibration frequency of perilous rocks on the slope and ensure the safety of technicians. At the same time, the pilot application is strengthened more to further verify and improve the early identification model of the perilous rock proposed in this paper.

### 5. Conclusions

(1) This paper takes the perilous rock on the soil slope as the research object, assuming that the foundation soil is homogeneous and isotropic, the damping ratio is less than 1, and the deformation is linear elastic deformation in the amplitude range. The vibration model of the perilous rock on the slope was simplified into a vibration model with multiple degrees of freedom. Based on the dynamic theory, the quantitative relationship among the natural vibration frequency, mass, side length, deformation modulus of the foundation soil, and buried depth of the perilous rock on the soil slope was established; that is, the dynamic characteristic model of the perilous rock on the slope was set up.

(2) The indoor model tests were carried out with models of different mass, different side lengths, and different buried depths. The experimental findings demonstrate that as the buried depth of perilous rock increases the frequency of its natural vibrations grows steadily in a quadratic fashion. The natural amplitude vibrations decrease quadratically as the mass of potentially perilous rock increases. Table 3 shows that the dynamic characteristic model of the perilous rock developed in this research is accurate, as predicted by the tests.

(3) Combined with the dynamic characteristic model and limit equilibrium model of the perilous rock on the soil slope, the evaluation model of perilous rock stability based on the natural vibration frequency was established, and the model was verified to be correct by a field test. Based on the research results, the buried depth of the perilous rock can be calculated indirectly by measuring its natural vibration frequency. Its safety coefficient on the soil slope can be calculated. In this way, these calculations offer the theoretical basis and technical support for early identification, stability evaluation, and monitoring and early warning.

**Author Contributions:** Y.J. and L.W. contributed to the conception of the study and performed the experiment; Y.J. and G.S. were responsible for data collection; Y.J., G.S. and T.J. made significant edits to the manuscript; Y.J., L.W. and G.S. were in charge of making important revisions to the manuscript; J.Z. and Z.L. helped perform the analysis with constructive discussions. All authors have read and agreed to the published version of the manuscript.

**Funding:** This research was funded by the National Natural Science Foundation of China (No. U1704243) and the Natural Science Foundation granted by the Department of Education, Anhui Province (No. KJ2020A0235).

**Institutional Review Board Statement:** Not applicable.

**Informed Consent Statement:** Not applicable.

**Data Availability Statement:** Not applicable.

**Conflicts of Interest:** The authors declare no conflict of interest.

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
