# Peer review of "Research on Stability Evaluation of Perilous Rock on Soil Slope Based on Natural Vibration Frequency"

_applsci, doi:10.3390/app13042406_

Round 1

Reviewer 1 Report

I see here a relatively good paper with some flaws. Some important revisions are needed prior to publication. See below:

1) Language usage is comprehensible but shorter sentences may help in the abstract, introduction and discussion, mainly. Please adapt them when you can.

2) Perilous rock instability should be better defined, not only operationally but also theoretically, to help non-familiar readers with geology to understand the processes at stake.

3) THe originality and novelty of the paper should be better clarified in the discussion section.

4) I don't see important comments on the direction of future research in the field. This is a flaw of the present paper. Readers are potentially interested in receiving an indication for future research rounds.

5) The policy implications of this study are only indirectly delineated. There is so much material stemming from the study that can you discuss in a policy perspective. We believe there is room to do so in the discussion section.

THank you.

Author Response

I would like to express my gratitude to editor and reviewer for reviewing the paper in their busy schedule. The paper has been revised according to the suggestions. Please review again. Looking forward to your reply.

  • Professional institutions have been invited to polish and revise the paper in English. Some long and difficult sentences have been revised as far as possible without changing their original meaning.
  • The article adds the theoretical definition of perilous rock instability.(page 1, line 32-36)
  • In the discussion part, the originality and novelty of the paper are clarified.(page 9, line 242-245)
  • Some comments on the future research directions in this field are added.(page 12, line 337-346)
  • The government's policies and attitudes in this research field are included in the paper.(page 12, line 335-337)

Reviewer 2 Report

First of all, the language used in this manuscript is VERY unreadable. So many long sentences make it difficult to read.

Major comments:

1. In the Abstract, the authors did NOT list the research objective clearly.

2. The authors used several terms to describe the "research topic", which makes it confusing.

3. In the Introduction, the research objective is still NOT clearly presented.

4. In the "Materials and Methods" part, the authors did NOT clearly describe the "Methods" used in this manuscript.

5. For the symbols used in the equations, some of them are MISSING the definitions.

6. In the "Model test" part, the authors did NOT present the "Model" clearly.

7. In the Figure 6, the authors did NOT present the slope angle, which is an important factor related to the perilous rock stability on the slope.

8. In Figure 8, some of the symbols of the factors are MISSING.

9. Even in the "Conclusion" part, the authors did NOT clearly present the methods, results, and highlights of this manuscript.

Author Response

I would like to express my gratitude to editor and reviewer for reviewing the paper in their busy schedule. The paper has been revised according to the suggestions. Please review again. Looking forward to your reply.

  • The article adds the research objective in the abstract.(page 1, line 12-14“The buried depth of the perilous rock, acting as a challenge to deal with, primarily determines its stability, and indirect rapid identification of its buried depth is the key to its stability evaluation. The paper aims to find a new and quick method to measure the buried depth of perilous rocks on soil slope, to solve the hard-to-measure buried depth in its stability evaluation.”)
  • The "research topic"has been changed to uniform terms.(page 1, line 20, line 22, line 24, line 26; page 2, line 69, line 90, line 97; page 3, line 103, line 122; page 4, line 140; page 8, line 229, line 230; page 11, line 307, line 308. dangerous rock, hazardous rock---perilous rock; burial depth---buried depth; slope---soil slope)
  • The paper adds some content in the introduction to show the objective of the study.(page 2, line 54-60. “The buried depth of the perilous rock on the soil slope is an important parameter to evaluate its stability, which is hard to measure directly by instruments, so it brings great challenges to its identification, stability evaluation, and monitoring and early warning. To solve this problem, this paper, through theoretical deduction, model tests, and field tests, constructs a method based on dynamic characteristic parameters to identify the buried depth of perilous rock on the soil slope, and then achieves its stability evaluation with limit equilibrium theory. ”)
  • In the "materials and methods" section, the "methods" used in this manuscript are further described.(page 3, line 112-119. “The perilous rock vibration model is simplified into a multi-freedom vibration model. The quantitative relationship, among perilous rock mass, rock bottom response coefficient, perilous rock buried depth, and natural horizontal vibration frequency, is established by theoretical inference. In the experiment, several perilous rock models with different masses and different buried depths were set up, and the vibration frequencies under the same buried depths and the buried depths under the same vibration frequencies were compared, researching the relationship between the buried depths of the perilous rock and the vibration frequencies. ”)
  • Some definitions of the symbols in the formulas have been added.(page 5, line 150ï¼›page 7, line 209. A , )
  • Two paragraphs in the article describe the model. (page 6, line 180-183ï¼›page 7, line 194-197. “Attach the vibration detector to the model's summit, then use epoxy glue to connect the detector's fixed plate to the perilous rock block shown in the foreground. Installation of the layout sensor can occur after the epoxy resin has dried and the test model has been taken entirely. ”; “In the experiment, cubes of different sizes and qualities were used to simulate perilous rocks. The dimensions of models A, B and C are 0.3m*0.3m*0.4m, 0.3m*0.3m*0.3m and 0.3m*0.3m*0.2m respectively, and the masses are 84.3 kg, 63.2 kg and 42.5 kg respectively. Figure 5 depicts the experiment's perilous rock model. ”)
  • The slope angle is added in Figure 6 and illustrated in the text. (Page 7,  line 209, line 215. “The slope angle  remains unchanged.”) 
  • Symbols for some factors are added in Figure 8. (page 9, line 268)
  • In the "Conclusion" section, the methods, results and highlights of this manuscript are re-described. (page 12, line 348-372. )

Round 2

Reviewer 1 Report

Nice and correct revision overall. I believe the manuscript is now rather standard and clear enough

Author Response

Thank you very much for giving us an opportunity to revise our manuscript and your positive and constructive comments and suggestions on our manuscript (Manuscript ID: applsci-2192753). The manuscript has been revised according to the suggestions. Please review again. Looking forward to your reply.

  • The correlation between model test and theoretical derivation and the necessity of model test are further elaborated.(page 5, line 167-169“In order to further verify the correctness of formula (17), the following part verifies whether the relationship between the buried depth, mass and vibration frequency of perilous rock is consistent with formula 17 through model test.”)
  • Moderate English changes have been made in some places.(page 11, line 321-322,line 327-334)

Reviewer 2 Report

Minor comments:

1. The authors need to have a thorough check typos in this manuscript.

2. Please check the format and font of ALL the figures, and make them constant.

3. Please check Figure-7. What does the blue and the black line mean? The black dots are missing or overlapped by the black curve ??

Author Response

Thank you very much for giving us an opportunity to revise our manuscript and your positive and constructive comments and suggestions on our manuscript (Manuscript ID: applsci-2192753). The manuscript has been revised according to the suggestions. Please review again. Looking forward to your reply.

  • We have thoroughly checked the typos in this manuscript.(page 1, line 9, line 30; page 3, line 121; page 12, line 361)
  • The format and font of allthe figures have been checked and adjusted.(page 3, line 119-120; page 4, line 128-129; page 6, line 191-194; page 7, line 201-202, line 218-219; page 8, line 231-232; page 9, line 271-272, page 11, line 321-322)
  • Figure 7 in the previously submitted manuscript has not been fully expanded and has been adjusted.(page 8, line 231-232)
  • Moderate English changes have been made in some places.(page 11, line 321-322, line 327-334)
